# Oral Health: Global Research Performance under Changing Regional Health Burdens

**DOI:** 10.3390/ijerph18115743

**Published:** 2021-05-27

**Authors:** Salim Ahmadi, Doris Klingelhöfer, Christina Erbe, Fabian Holzgreve, David A. Groneberg, Daniela Ohlendorf

**Affiliations:** 1Institute of Occupational Medicine, Social Medicine and Environmental Medicine, Goethe-University, 60590 Frankfurt am Main, Germany; ahmadi.salim@rocketmail.com (S.A.); klingelhoefer@med.uni-frankfurt.de (D.K.); occup-med@uni-frankfurt.de (D.A.G.); ohlendorf@med.uni-frankfurt.de (D.O.); 2Department of Orthodontics, School of Dentistry, University Medical Centre of the Johannes Gutenberg University, 55131 Mainz, Germany; erbe@uni-mainz.de

**Keywords:** oral health, oral health burden, bibliometrics, publication output, socio-economic influences

## Abstract

*Objectives:* Inadequate oral hygiene still leads to many serious diseases all over the world. Therefore, this study aimed to analyze scientific research in the field of oral health in order to be able to comprehend their relevant subject areas, research connections, or developments. *Methods***:** This study aimed to assess the global publication output on oral hygiene to create a world map that provides background information on key players, trends, and incentives of research. For this purpose, established bibliometric parameters were combined with state-of-the-art visualization techniques. *Results***:** This study shows the actual key players of research on oral hygiene in high-income economies with only marginal participation from lower economies. This still corresponds to the current burden situations, but they are more and more shifting to the disadvantage of the low-income countries. There is a clear North–South and West–East gradient, with the USA and the Western European nations being the most publishing nations on oral hygiene. As an emerging country, Brazil plays a role in the research. *Conclusions***:** The scientific power players were concentrated in high-income countries. However, the changing epidemiological situation requires a different scientific approach to oral hygiene. This requires an expansion of the international network to meet the demands of future global oral health burdens, which are mainly related to oral hygiene.

## 1. Introduction

The purpose of oral hygiene is to prevent the formation of plaque, which indexes caries, gingivitis, and periodontitis [1]. The reduction of pathogenic microbiota in the oral cavity is executed by supervised dental brushing and the use of oral hygiene specific aids, such as antibacterial mouthwash and topical fluoride application [2,3].

Oral health refers to the freedom from inflammation and discomfort of the tooth, mouth, and jaw system and is therefore a critical part of general health and quality of life [4].

The Global Burden of Disease Study 2017 estimated that oral diseases affect 3.5 billion people worldwide [5]. According to this, around 60–90% of schoolchildren and almost 100% of adults worldwide suffer from carious lesions [4]. Serious periodontal diseases, which can lead to tooth loss, are found in 15–20% of adults aged 35–44. According to an increase in prevalence, around 30% of people aged between 65 and 74 even exhibit a complete tooth loss [6]. Additionally, unhealthy nutrition or tobacco consumption can lead to considerable carious lesions and to the development of enamel defects, tooth erosion, or periodontal disease [4].

Still, most people in developing countries do not have access to oral health programs [5]. Nevertheless, poor oral health is a problem not only in poorer or less developed countries but also in high-income countries, where an inverse correlation of the social gradient in terms of social status and caries has been demonstrated [6].

The focus of scientific research in the field of oral hygiene is on oral disease prevention and reduction of the burden on health systems, which is estimated as 5% [6]. In addition to increasing knowledge on caries prevention [7,8,9,10,11] and demographic changes [12,13], the economic importance of oral care products is considered as the driving force behind research and development (R&D) on oral health.

Since poor oral hygiene as a key factor for oral health still causes suffering for many people worldwide and is additionally associated with high financial societal burdens, it is necessary to take an in-depth look at relevant research efforts. Therefore, this study aimed to conduct a thorough evaluation of the research from chronological, geographical, and socio-economic aspects. The results thus provide background information for future planning of research programs, targeted promotions, and meaningful measures to increase individual awareness of the importance of good oral hygiene.

## 2. Methods

### 2.1. Methodological Platform

The applied methods are embedded in a methodological platform that combines established bibliometric approaches with advanced parameters and the inclusion of density equalizing map projections (DEMP [14]) for the first time. The New Quality and Quantity Indices in Science (NewQIS) platform [15] was established to analyze the global research output of scientific topics in order to provide useful background information in respective research to all parties involved, all for better future planning.

### 2.2. Search Source and Strategy

The core collection of the online database Web of Science (WoS) served as the most suitable data source because it also provides citation figures via its Journal Citation Report (JCR).

The date of the retrieval of data was 10 June 2016. In order to include only full years in the analyses, the evaluation period was set from 1900 to 2015.

The following string of search terms was searched in the title of the articles: ((mouth OR teeth OR dental OR plaque OR oral) AND (clean* OR hygiene* OR rinse OR wash* OR brush*)) OR toothbrush* OR toothpaste* OR “dental floss*” OR “dental silk” OR “plaque removal” OR “interdental brush*” OR dentifrices OR “plaque removal” OR ((caries OR periodontitis OR gingivitis OR tartar) AND (preventi* OR prophylaxis*) OR [(fluoride* AND (varnish* OR “mouth rins*”)] NOT TITLE: (“plaque indices” OR nourishment* OR alimentation OR “eating habits” OR “eating behavior” OR diet OR nutrition* OR “prostheses cleaning” OR airscaler).

This complex search term ensured that articles on cleaning methods, cleaning tools, and cleaning instruments were found but not articles on plaque indices, dental instruments, and nutritional behavior. Subsequent filtering for original articles as the document type meant that only original studies were included in the analysis. The representativeness of the included articles was ensured by extensive random checking of the article list.

### 2.3. Data Processing, Analyses, and Visualization

The bibliometric meta data of the identified articles were downloaded from the WoS. Data were then sorted according to the various analysis parameters and stored in a Microsoft Access database. Some analysis categories, such as institutions, had to be manually standardized because different values are used.

The subsequent generated database was analyzed according to a variety of parameters, such as time of publication, country of origin, number of citations, citation rate, publication source, and research field. Additionally, country-specific data were analyzed in terms of their socio-economic characteristics (population in million inhabitants, gross domestic product (GDP) in USD 1000 billion) [16]. For this purpose, we calculated the quotients from the number of articles and the socio-economic indicators of the publishing countries (R_POP_, R_GDP_).

Correlations between publication year and publication numbers or citation numbers were calculated using Spearman correlation.

Geographical results were visualized based on the established DEMP technique by Gastner and Newman using the ArcGIS application [1]. This procedure distorts the country sizes according to the value of the parameter to be analyzed based on the physical principle of density equalization. As a result, countries with low values are reduced in size and countries with high values are enlarged. The oceans and Antarctica are thereby given medium values in order to preserve the general structure of the continents [17].

International collaborations were depicted as networking diagrams represented by nodes and connecting lines of varying strength.

## 3. Results

In 115 years, a total of 4812 oral hygiene-specific articles (*n*) were published in the WoS.

### 3.1. Chronological Analyses

After the first article was published in 1902, a significant increase in the number of scientific publications was observed until 2015 (Spearman correlation coefficient, *p* < 0.01). The average number of studies increased from *n* < 100 per year until 2001 to *n* < 200 until 2010 and subsequently raised to >200 after 2011, with quantitative peaks in 1964 and the 1980s. The number of citations (c) increased continuously from c > 100 in 1961 to c > 1000 in 1971 and c > 2000 after 2000 (*p* < 0.001). In 1946 (*n* = 131), 1950 (*n* = 116), and 1964 (*n* = 1014), the number of annual citations was above the quantitative average. The maximum value (c = 2235) was reached in 2009, followed by a steady decline in the annual citation frequency (Figure 1).

The most frequently cited articles on oral hygiene are shown in Table 1.

### 3.2. Geographical Analyses

Due to missing country of origin information, the number of articles for geographical analyses decreased to *n* = 4350.

The USA published the largest number of articles on oral hygiene with *n* = 1274 (29.9%), followed by the UK (*n* = 634, 14.6%), Germany (*n* = 361, 8.3%), Spain (*n* = 268, 6.2%), and Israel (*n* = 261, 6%). Outside North America and Europe, Brazil (*n* = 248, 5.7%) produced the most oral hygiene articles (Figure 2A).

Country-specific citation analysis indicated a leading position of the USA (c = 18,536), followed by the UK (c = 10,451), Sweden (c = 6680), and Germany (c = 3920) (Figure 2B).

Additionally, the average number of citations per article (citation rate = cr) of the countries with at least 30 articles on oral hygiene (threshold) was calculated. Denmark (cr = 25.9) showed the highest value, followed by Sweden (cr = 24.2), Norway (cr = 20.1), New Zealand (cr = 19.0), and Canada (cr = 17.2) (Figure 2C).

In terms of socio-economic analyses, the number of country-specific articles was related to the number of countries’ population in millions (R_POP_) [16]. Here, the Scandinavian countries dominated: Sweden (R_POP_ = 28.4), followed by Finland (R_POP_ = 26.4), Norway (R_POP_ = 22.5), and Denmark (R_POP_ = 18.1). Germany and the USA, countries with high publication numbers, dropped to position 12 (R_POP_ = 4.4) and 14 (R_POP_ = 3.9) (Figure 3A).

When the number of country-specific publications was related to the gross domestic product (GDP) in USD 1000 billion [16] (Table 2) (R_GDP_), the Scandinavian countries again dominated. Finland (R_GDP_ = 512.5) was ahead of Sweden (R_GDP_ = 484.1), Denmark (R_GDP_ = 296.3), Norway (R_GDP_ = 231.9), and the UK (R_GDP_ = 215.2). The high productive countries Germany and USA only ranked 13th (R_GDP_ = 93.5) and 19th (R_GDP_ = 73.1) (Figure 3B).

In line with the highest number of published articles, the USA was the country with the most collaborations (*n* = 282; 44.8%).

The UK published 137 collaborative articles, followed by Germany (*n* = 113), Sweden (*n* = 58), and Brazil (*n* = 54). The most productive international collaboration was identified between the USA and the UK, with 45 articles. In addition, the USA collaborated with Germany and Canada on more than 30 joint articles. Overall, the most active international collaborations were established with the USA. For Germany, the USA remained the main cooperation partner, next to the Netherlands, Switzerland, and the UK (Figure 4).

Most articles (*n* = 3265) were assigned to the subject area of dentistry, oral surgery, and medicine; followed by public, environmental, and occupational Health (*n* = 491); general and internal medicine (*n* = 195); pediatrics (*n* = 190), and chemistry (*n* = 118).

Chronological analyses identified dentistry, oral surgery, and medicine as the leading subject category throughout the examination period (Figure 5).

## 4. Discussion

The aim of the present study was to analyze and interpret the global publication output on oral health according to various parameters. The results showed a very differentiated pattern.

A total of 4812 articles on oral hygiene were published in WoS, of which 4350 articles could be used for the geographic analyses. The publication numbers as well as the number of citations were shown to increase significantly over time. The article count reached a peak in 1964 (Figure 1) when the term “oral hygiene” was coined in the most frequently cited article of this study, “Simplified Oral Hygiene Index”, by Green & Vermillion [18]. Marketing of the first electric toothbrush in 1960 [19] accounted for further technical development and a remarkable rise in article numbers.

Resembling the growing number of publications, the citation count reached a peak in 1978 when the second most cited article on the effect of controlled oral hygiene by Axellson and Lindhe [20] was published. In the early 1980s, oral hygiene research gained increasing popularity. At this time, a link between technical developments concerning electric toothbrushes, epidemiological studies, and publication numbers was determined. By 2007, the annual volume of publications had multiplied more than 3 times. Four of the ten most cited publications were published between 1991 and 2000 and dealt with long-term studies on caries and periodontal diseases, antibiotic mouth rinses, and caries prevention (Table 1). After 2011, annual research productivity increased to more than 200 papers concerning technical development and efficacy of methods such as the oral irrigator in reduction of bleeding, gingivitis, and plaque compared with dental floss [21].

The highest number of publications was reached in 2015. The maximum number of citations, however, was already recorded in 2004. The continuous decrease in citation numbers after 2009 is due to the cited half-life (citation half-life, CHL) [22], as the more recent publications have not yet had enough time to generate the maximum of their citations. When evaluating at a later date, these numbers will certainly increase.

When country-specific oral hygiene research was analyzed, the USA (*n* = 1274 out of 4812 articles) was identified as the most productive nation, followed by the UK (*n* = 634) and Germany (*n* = 361). The USA had previously been nominated as the most productive nation of dental publications [23]. This is also due to their increased funding for R&D. According to this, the National Institute of Health (NIH) funding of dental research was estimated nearly USD 140,000,000 in 2017 [24].

As the only economy not classified as a high-income country, the South American country of Brazil played a significant role in oral hygiene research, with 248 publications. This is certainly caused by the development of Brazilian health research, based on the National Strategy for Science, Technology, and Innovation. This includes a strong increase in postgraduate programs in the field of dentistry, which in turn extended the number of published studies [25].

After steadily increasing the annual R&D efforts, Brazil ranked 10th in the international comparison of expenditure on R&D in 2017 [26]. However, the presence of Brazilian publications, measured by citations, did not resemble the quantitative increase, presumably due to the selection of journals with a lower impact factor.

Still, people in developing countries are exceedingly affected by dental caries [8] and periodontal disease [27], as well as tooth loss [28], while their populations do not benefit from preventive oral health programs [29]. A global strengthening of oral health programs is urgently needed to prevent oral diseases, which would promote the integration of developing countries into global research [29]. However, economic growth and thus the development of low- or middle-income countries could indirectly lead to a stricter segregation of population groups through the chronic effects of poor oral hygiene, such as dental disease and tooth loss [4].

Overall, scientific output was higher in North America and Europe than in Asia, Africa, and South America, consistent with the bibliometric study on dental research by Gil-Montoya et al. [30]. The results of this study reflect the hypothesis that English-speaking countries are leading in the literature comparison, since non-English literature from other countries of origin is not or is only included to a small extent in the databases used.

In this study, the USA, the UK, Sweden, and Germany dominated the ranking in terms of citation count. These findings coincide with other studies where the USA, the UK, Japan, and Germany were determined to be the most productive nations concerning dentistry, oral surgery, and medicine [30]. Furthermore, these countries are the representative markets for consumer research in oral health. The number of citations correlated with the quantity of studies in the USA, the UK, and Germany, as stated before [25]. However, a previous study found a falsification of results in the USA due to up to 30% self-citation [23].

Similar to Cabrini-Gracio et al. [25], the highest average citation rate was recorded in the Scandinavian countries of Denmark, Sweden, and Norway. This finding is based on the comprehensive caries prevention measures of these countries and on the fact that public health data is publicly viewable and transparent in Scandinavian countries, which leads to numerous publications in high-ranking journals. To define the commitment of single countries in oral hygiene research, the scientific output in terms of socio-economic abilities was determined. Highest ratios, relative to both GDP and population, were recorded in the Scandinavian countries. These results correlate with those from previous bibliometric studies [30] and may be explained by the Scandinavian systems’ intervention strategies for caries reduction, such as free dental care for children and adolescents and a special dental care program for adults with reduced mobility or a reduced physical or mental constitution [31].

The high citation rate of Danish publications is mainly attributable to three studies that were placed in the ranking of the 15 most frequently cited articles. These studies dealt with cariostatic measures by fluoride [32], the effect of tooth brushing and plaque reduction [33], and bacteremia after oral surgery in periodontal disease [34].

Publication performances of Sweden were strengthened by four highly cited papers by Axelsson and Lindhe [20,35,36,37] on epidemiologic studies concerning the effect of oral hygiene on oral diseases, which were among the 10 most cited publications (Table 1).

The intensive research, which is reflected in the quantity and quality of the studies, in cooperation with national interventions such as the Swedish Quality Register for Caries and Periodontal Diseases (SkaPa) for systematic evaluation of oral health and quality of dental care, has led to the Scandinavian countries being among the “very low- and low-caries countries” [3].

Most of the collaborative articles were produced by the UK universities Cardiff University, University of Bristol, and University of Wales, in collaboration with various US institutes. This research mainly focused on the caries-preventive effect of oral hygiene products such as toothpastes and mouth rinses [38]. The almost as intensive partnership between USA and Canada resulted in joint publications on new technical developments in the field of oral hygiene products between BioSci Research Canada Ldt. and the USA’s Oral-B Laboratories [39].

The intensive research of the German Braun AG in Kronberg with the Optiva Corp. in the USA and UK accounted for studies on plaque reduction by electric toothbrushes and therefore contributed significantly to the technical development of oral hygiene [40].

In summary, a regional concentration of articles was observed, with only five nations being responsible for more than 50% of the article production.

Although broad international networking already exists, future research planning should be expanded to include currently underrepresented regions. Mutual benefit can be achieved through the exchange of knowledge and new approaches, especially with regard to education and improved awareness of oral hygiene.

The results presented may be somewhat biased by the composition of the generated database due to the characteristics of the online database WoS, which is known to have a bias towards English articles. In addition, not all articles can be included in the analysis due to WoS’s stringent requirements for indexed journals. The title search strategy applied also reduced the size of the database. Another limitation that must be considered when interpreting the results is the correctness of the error-prone citation parameters.

Nevertheless, the advantages of the database as one of the most used sources worldwide and the quality of the listed articles combined with the guaranteed representativeness of the search strategy justify this approach.

## 5. Conclusions

The present study was conducted to obtain information on the global research output on oral hygiene. It was found that the current major players in oral hygiene research are high-income economies. Only marginal involvement of lower-income economies can be observed. Thus, the commitment of research efforts is clearly concentrated and corresponds to the existing high burden countries. Nevertheless, the situation seems to be changing with the adoption in the developing countries of the lifestyle of the industrialized countries, and thus health burdens due to poor nutrition will become apparent there as well. Future research must take these circumstances into account. This requires the expansion of the international research network, which should include these regions, so that appropriate measures can be taken there as well, based on sound science. These requirements should be taken into account by all stakeholders. Scientists, planners, funders, and stakeholders involved in science are therefore encouraged to align their research efforts with this changing situation in mind.

## Figures and Tables

**Figure 1 ijerph-18-05743-f001:**
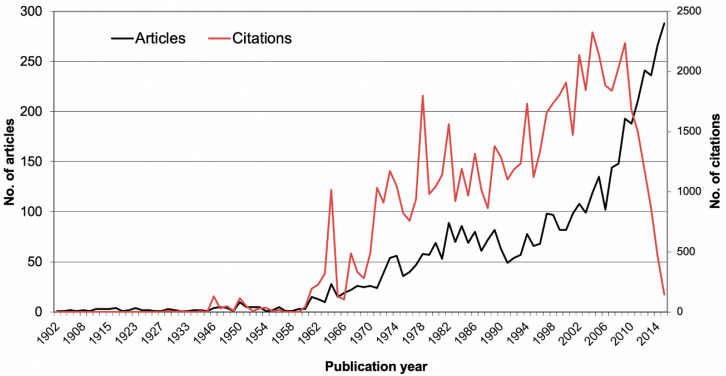
Number of articles and citations over time.

**Figure 2 ijerph-18-05743-f002:**
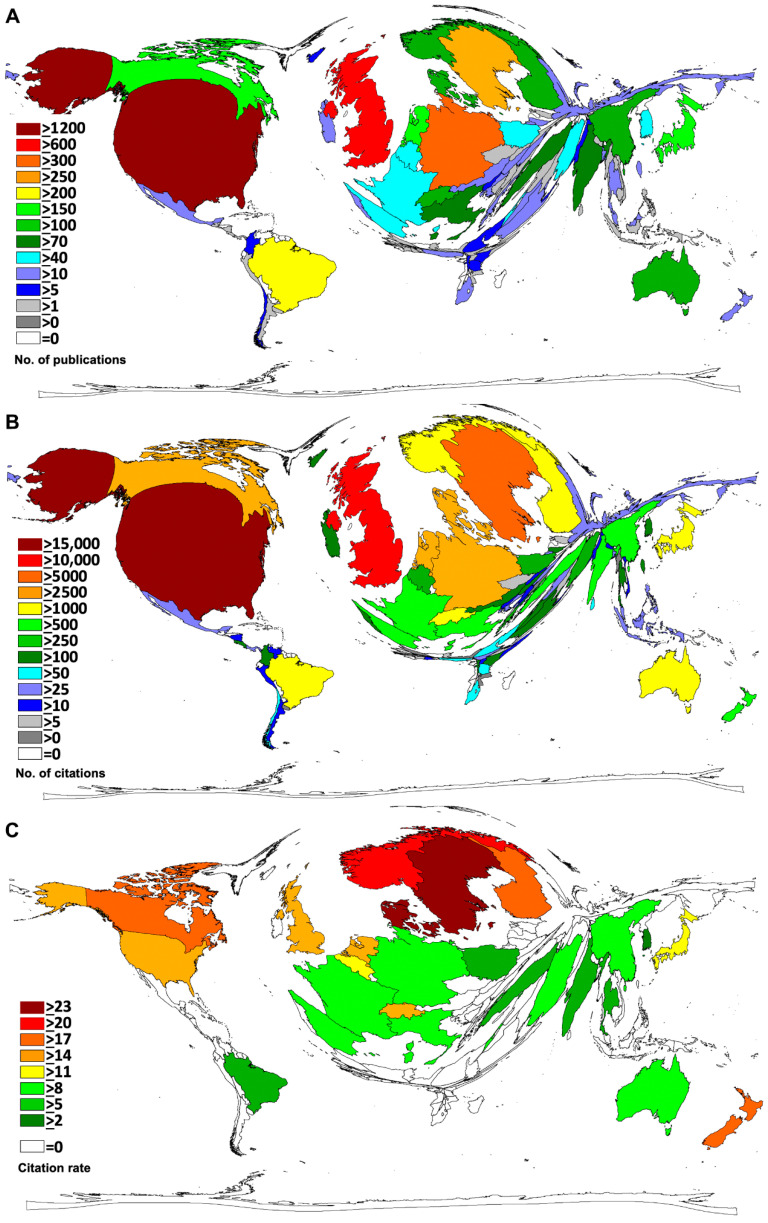
Countries’ performances on oral hygiene. (**A**) Number of articles. (**B**) Number of citations. (**C**) Citation rate (threshold > 30 articles).

**Figure 3 ijerph-18-05743-f003:**
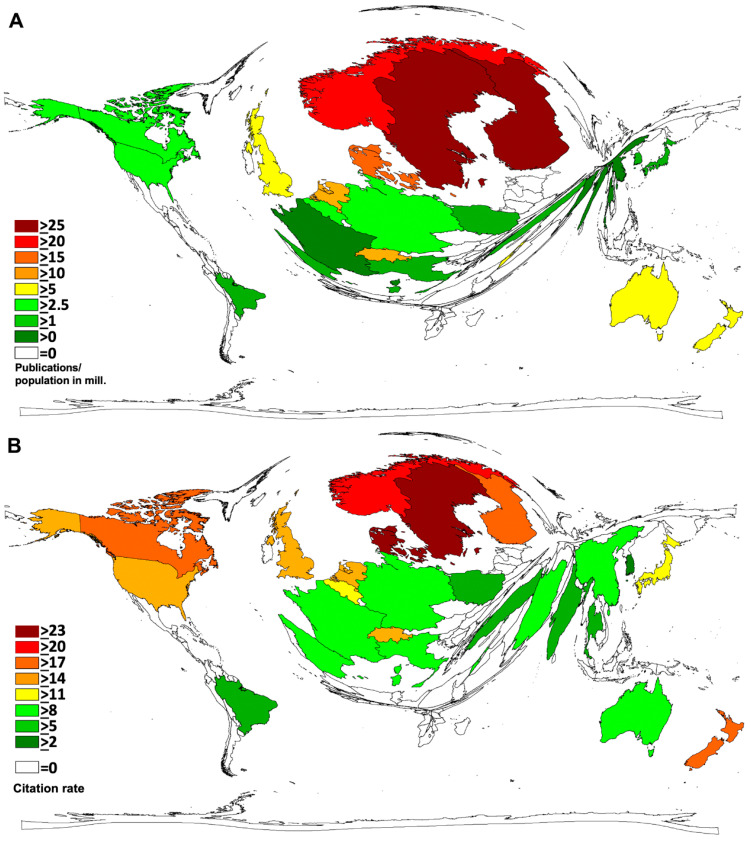
Socio-economic parameters. (**A**) R_POP_ = number of articles per population in millions [16] (threshold value: >30 articles). (**B**) R_GDP_ = number of items per gross domestic product (GDP) in USD 1000 billion [16] (threshold value > 30 articles).

**Figure 4 ijerph-18-05743-f004:**
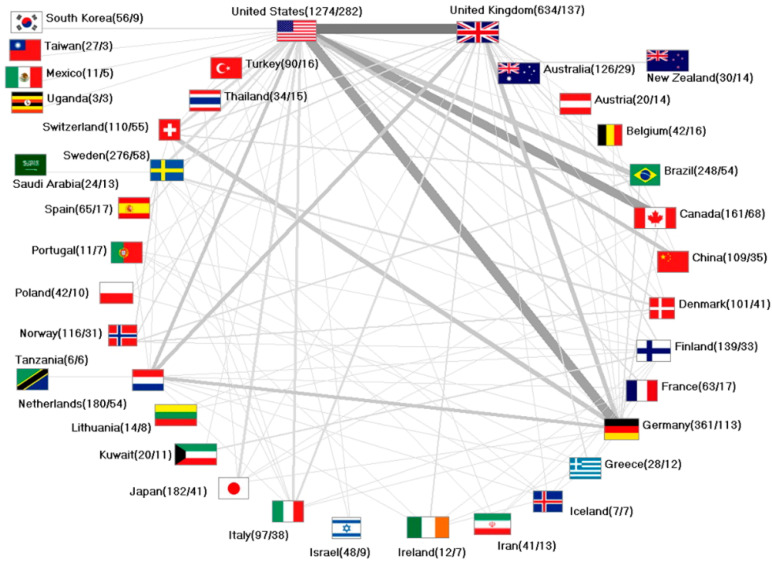
International network on oral hygiene. Numbers in brackets = number of articles/number of cooperation articles.

**Figure 5 ijerph-18-05743-f005:**
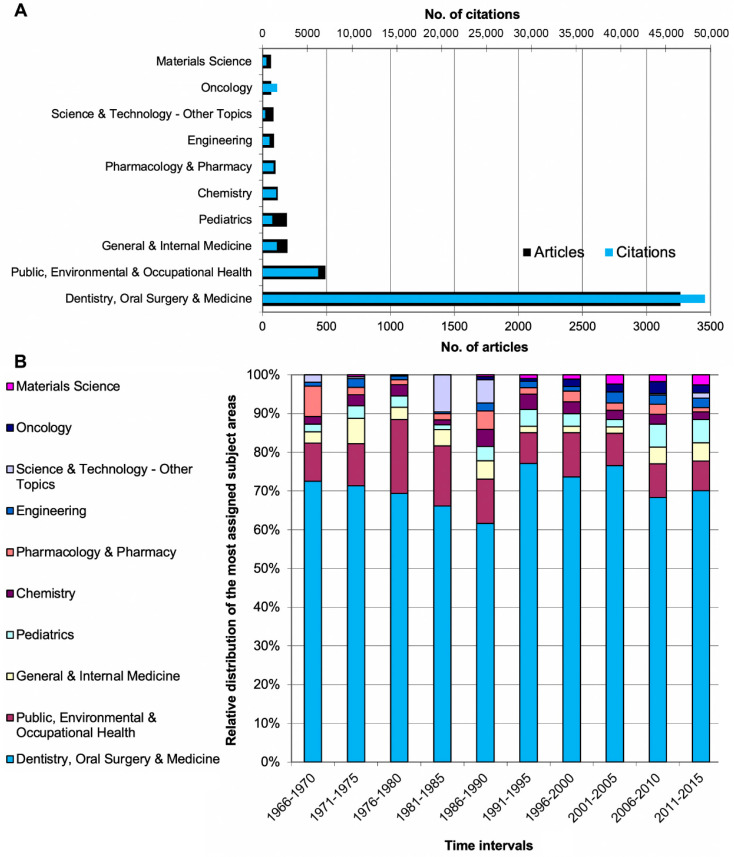
Subject areas of Web of Science categories. (**A**) Ten most frequently assigned areas with number of articles and citations. (**B**) Relative distribution of the most frequently assigned subject areas in 5-year intervals from 1966–2015.

**Table 1 ijerph-18-05743-t001:** Most cited articles.

Author	Country	Year	Citations	Title	Journal
Greene, Vermillion	USA	1964	646	Simplified oral hygiene Index	The Journal of the American Dental Association
Axelsson, Lindhe	Sweden	1978	375	Effect of controlled oral hygiene procedures on caries and periontal-disease in adults	Journal of Clinical Periodontology
Suomi et al.	USA	1971	308	Effect of controlled oral hygiene procedures on progression of peridontal disease in adults—results after third and final year	Journal of Clinical Periodontology
Featherstone	USA	2000	297	The science and practice of caries prevention	Journal of American Dental Association
Featherstone	USA	1999	276	Prevention and reversal of dental caries: role of low level fluoride	Community Dentistry and Oral Epidemiology
Axelsson, Lindhe	Sweden	1981	266	Effect of controlled oral hygiene procedures on caries and peridontal-disease in adults—results after 6 years	Journal of Clinical Periodontology
Axelsson, Lindhe	Sweden	1974	241	Effect of a preventive programme on dental plaque, gingivits and caries in schoolchildren—results after one and 2 years	Journal of Clinical Periodontology
Axelsson, Lindhe	Sweden	1991	225	On the prevention of caries and peridontal-disease—results of a 15-year longitudinal-study in adults	Journal of Clinical Periodontology
DeRiso et al.	USA	1996	215	Chlorhexidine gluconate 0.12% oral rinse reduces the incidence of total nosocomial respiratory infection and nonprophylactic systemic antibiotic use in patients undergoing heart surgery	Chest
Sönju, Rölla	Norway	1973	209	Chemical analysis of acquired pellicle formed in 2 h on cleaned human teeth in-vivo—rato of formation and amino-acid analysis	Caries research

**Table 2 ijerph-18-05743-t002:** Socio-economic analyses (threshold > 30 articles). *n* = number of articles; GDP = gross domestic product [16]; R_GDP_ = ratio of number of articles and GDP in USD 1000 bn; R_POP_ = ratio of number of articles and population in mill. [16], sorted by R_POP_.

Country	*n*	GDP in 1000 bn US-Dollar	Population in Mill.	R_GPD_	R_POP_
Sweden	276	570.1	9.72	484.13	28.40
Finland	139	271.2	5.26	512.54	26.43
Norway	116	500.2	5.14	231.91	22.57
Denmark	101	340.8	5.56	296.36	18.17
Switzerland	110	712.1	8.06	154.47	13.65
Netherlands	180	866.4	16.87	207.76	10.67
UK	634	2945	63.74	215.28	9.95
New Zealand	30	198.1	4.4	151.44	6.82
Israel	48	303.8	7.82	158.00	6.14
Australia	126	1444	22.5	87.26	5.60
Canada	161	1789	34.83	89.99	4.62
Germany	361	3860	80.99	93.52	4.46
Belgium	42	534.7	10.44	78.55	4.02
USA	1274	17,420	318.9	73.13	3.99
Italy	97	2148	61.68	45.16	1.57
Japan	182	4616	127.1	39.43	1.43
Spain	65	1407	47.73	46.20	1.36
Brazil	248	2353	202.6	105.40	1.22
South Korea	56	1410	49.03	39.72	1.14
Turkey	90	806.1	81.61	111.65	1.10
Poland	42	546.6	38.34	76.84	1.10
France	63	2847	66.25	22.13	0.95
Iran	41	404.1	80.84	101.46	0.51
Thailand	34	373.8	67.74	90.96	0.50
China	109	10,380	1355.7	10.50	0.08
India	84	2050	1236.3	40.98	0.07

## Data Availability

All data are available via the corresponding author.

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
