# Peer review of "Oral Health: Global Research Performance under Changing Regional Health Burdens"

_ijerph, 2021, doi:10.3390/ijerph18115743_

Round 1

Reviewer 1 Report

Dear Authors,

Congratulations on the successful completion and submission of your research manuscript for the peer-review process.

The results of the study are well-presented.

Please take note of the following corrections that require your kind attention:

Line 165, Line 191: 4.812 & $14.000.000 - is it 4,812 articles? please use a comma to express the correct values.

Kindly include a paragraph under discussion explaining the rationale of your study, limitations of your study, and future perspectives.

Can 'Oral Health Knowledge / Awareness', 'Attitude and behavior towards Oral Health/care' of people across different geographical locations also be included as significant parameters when assessing the global research performance on Oral Health?

Thank you.

Author Response

Reviewer 1

General comment:

Congratulations on the successful completion and submission of your research manuscript for the peer-review process. The results of the study are well-presented.

Response: We thank the reviewer for this overall assessment of our study.

Comments:

Line 165, Line 191: 4.812 & $14.000.000 - is it 4,812 articles? please use a comma to express the correct values.

Response: We have corrected all numbers accordingly throughout the entire text. For this, we have deleted the dots for all four-digit numbers.

Kindly include a paragraph under discussion explaining the rationale of your study, limitations of your study, and future perspectives.

Response: We included corresponding paragraphs at the beginning and at the end of the discussion.

Can 'Oral Health Knowledge / Awareness', 'Attitude and behavior towards Oral Health/care' of people across different geographical locations also be included as significant parameters when assessing the global research performance on Oral Health?

Response: This is certainly an interesting point. Unfortunately, corresponding data for an overall analysis that includes all nations is not available. Therefore, a corresponding analysis was not possible.

Reviewer 2 Report

Dear authors,

The present manuscript entitled "Oral Health: Global research performance under changing regional health burdens" is interesting research. However, the present reviewer has some concerns about that. 

Introduction

It was not possible to find the study aims.

Material and Methods

The study design should be clear and well written.

It was not possible to understand how you prepared the research.

Softwares? Statistical analysis?

The authors need to improve this section.

Results

The results are fascinating. However, it is not possible to figure out how it was produced.

Be careful with the abbreviations

Line 131 - citation missed.

You have many results and not methods described for those.

It's possible only to find the citation of figure 1 in the main text. 

The authors did not prepare a socio-economic evaluation as was described in the text. An analysis like that is higher than was described in table 2.

Discussion

Please, include the challenges to prepare for the present research and the limitations.

Conclusion

The authors have a considerable conclusion without an objective in the text.

Please, answer the study aims in conclusion.

The authors should be conscious and direct to write that.

Regards,

#Reviewer

Author Response

Comments:

Introduction

It was not possible to find the study aims.

Response: We apologize for this and have added a paragraph to the introduction section accordingly.

Material and Methods

The study design should be clear and well written.

It was not possible to understand how you prepared the research.

Softwares? Statistical analysis?

The authors need to improve this section.

Response: We have rewritten the methods section and added important information that was missing before.

Results

The results are fascinating.

Response: We thank the reviewer for this assessment of the results presented.

However, it is not possible to figure out how it was produced.

Response: Please see above. We have rewritten the methods section and added important information that was missing before.

Be careful with the abbreviations

Response: We have reviewed the abbreviations throughout the text and adjusted them accordingly.

Line 131 - citation missed.

Response: We added the citation accordingly.

You have many results and not methods described for those.

Response: Please see above.

It's possible only to find the citation of figure 1 in the main text.

Response: Sorry, it is indeed missing. We have supplemented the cross-reference accordingly.

The authors did not prepare a socio-economic evaluation as was described in the text. An analysis like that is higher than was described in table 2.

Response: We analyzed publication output by including socioeconomic indicators such as total population and gross domestic product of the countries. For this purpose, we calculated quotients from the number of articles and the socioeconomic indicators. This was now made clear in the methods section.

Discussion

Please, include the challenges to prepare for the present research and the limitations.

Response: We have added the corresponding paragraphs in the discussion.

Conclusion

The authors have a considerable conclusion without an objective in the text.

Please, answer the study aims in conclusion.

Response: We added a sentence to the conclusions accordingly.

Reviewer 3 Report

Overview

This original article aims to assess the global publication output on oral hygiene and provides information on key players and  research trends in this field.The results are interesting although they could be further explored

There are several language errors throughout the text that need to be corrected

There are numerous bibliographic references missing throughout the text

Methods

When was the search carried out? Was it done by just one person? Was it done in duplicate?

Why was dental plaster duration included in the search strategy?

In the text the reference to Great Britain or the United Kingdom should be standardized, opting for the second form

Results

What was the source used for the population of the different countries? Regarding what year? The indicated population for several countries like the USA and India is below the current population.The same applies to GDP

Discussion

Publication dates are incorrect (line 165). In line 171 the dates are repeated.

This sentence needs to be clarified: “The continuous decrease in citations after 2009 may be either due to a loss of effectiveness of oral hygiene ?? or the cited half-life (citation half-life, CHL)21 

The amount of the investment in research is out of date (values refer to the year 2017)

It would also have been interesting to evaluate research productivity in view of the number of dental schools in each country

The limitations and weaknesses of this study are rarely discussed

Author Response

General comment:

This original article aims to assess the global publication output on oral hygiene and provides information on key players and research trends in this field. The results are interesting although they could be further explored.

Response: We thank the reviewer for this assessment of the results presented in this article.

Comments:

There are several language errors throughout the text that need to be corrected.

Response: We apologize for the mistakes made and have corrected many sentences in the text. We hope that the language is now suitable for publication.

There are numerous bibliographic references missing throughout the text

Response: We have added necessary references in many places in the text.

Methods

When was the search carried out? Was it done by just one person? Was it done in duplicate?

Response: We have added missing information accordingly in the method section. The search term was developed by the co-authors of the articles so that all necessary terms are included in the strategy.

Why was dental plaster duration included in the search strategy?

Response: The inclusion of this term does indeed seem somewhat irritating. The term was intended as a synonym for toothbrushing time. We checked the found entries according to the articles related to this term and found that this term did not provide any additional entries. Therefore, we were able to delete it from the search string to avoid irritation.

In the text the reference to Great Britain or the United Kingdom should be standardized, opting for the second form

Response: We have standardized the different terms for the UK.

Results

What was the source used for the population of the different countries? Regarding what year? The indicated population for several countries like the USA and India is below the current population. The same applies to GDP.

Response: We have added the World Factbook as a requested source and indicated the period as 2017. Therefore, the current data differ somewhat. The 2017 dataset was the most recent, which was fully available at the time of data collection.

Discussion

Publication dates are incorrect (line 165). In line 171 the dates are repeated.

Response: A misleading wording was indeed chosen here. The lower number of 4350 articles refers only to the geographical analyses due to missing information of the country of origin and not to the total number of entries found. We have added this information to the geographical results and to the discussion.

This sentence needs to be clarified: “The continuous decrease in citations after 2009 may be either due to a loss of effectiveness of oral hygiene ?? or the cited half-life (citation half-life, CHL)21 

Response: We apologize for the incorrect and misleading wording. The sentence has been rewritten.

The amount of the investment in research is out of date (values refer to the year 2017)

Response: The socioeconomic data included refer to 2017 values, as this dataset is the most recent fully available for all countries from UNESCO at the time of analysis.

It would also have been interesting to evaluate research productivity in view of the number of dental schools in each country

Response: This is a very interesting aspect. Unfortunately, these data are not available for all countries for this purpose. Therefore, comparison and evaluation would be hampered by incomplete data and non-valid evaluability.

The limitations and weaknesses of this study are rarely discussed

Response: We have added a paragraph discussing the limitations of the methods used.

Round 2

Reviewer 2 Report

Dear authors,

The present version sounds more like a scientific paper than the first one.

After minor reviews from the editorial team, the present manuscript has merit for publication.

Regards,

Reviewer